# The Pathogenic Role of C-Reactive Protein in Diabetes-Linked Unstable Atherosclerosis

**DOI:** 10.3390/ijms26146855

**Published:** 2025-07-17

**Authors:** Melania Sibianu, Mark Slevin

**Affiliations:** 1Doctoral School of Medicine and Pharmacy, George Emil Palade University of Medicine, Pharmacy, Science and Technology of Targu Mures, 38th Gh. Marinescu Street, 540139 Târgu Mureş, Romania; melania.sibianu@gmail.com; 2Center for Advanced Medical and Pharmaceutical Research, George Emil Palade University of Medicine, Pharmacy, Science and Technology of Târgu Mureș, 38th Gh. Marinescu Street, 540139 Târgu Mureş, Romania

**Keywords:** CRP, monomeric-CRP, inflammation, atherosclerosis, diabetes, AGE-RAGE

## Abstract

C-reactive protein (CRP) has long been recognized as a biomarker of systemic inflammation and cardiovascular disease (CVD) risk. However, emerging evidence highlights the distinct and potent pro-inflammatory role of its monomeric form (mCRP), which is predominantly tissue-bound and directly implicated in vascular injury and plaque destabilization. This narrative review explores the interactions and overlapping pathways that converge within and modulate CRP, mCRP, the associated pathophysiology of diabetes mellitus, and cardiovascular disease. We examine how mCRP promotes endothelial dysfunction, leukocyte recruitment, platelet activation, and macrophage polarization, thereby contributing to the formation of unstable atherosclerotic plaques. Furthermore, we discuss the critical influence of diabetes in amplifying mCRP’s pathogenic effects through metabolic dysregulation, chronic hyperglycemia, and enhanced formation of advanced glycation end products (AGEs). The synergistic interaction of mCRP with the AGE-receptor for AGE (RAGE) axis exacerbates oxidative stress and vascular inflammation, accelerating atherosclerosis progression and increasing cardiovascular risk in diabetic patients. Understanding these mechanistic pathways implicates mCRP as both a biomarker and therapeutic target, particularly in the context of diabetes-associated CVD. This review highlights the need for further research into targeted interventions that disrupt the mCRP-[AGE-RAGE] inflammatory cycle to reduce plaque instability and improve cardiovascular outcomes in high-risk populations.

## 1. Introduction

### 1.1. Atherosclerosis and Development of Unstable Plaques

Atherosclerosis remains a major global concern as the leading cause of death due to blood flow interruption. Due to the silent character of this condition, many people present sudden death without any prior symptoms, despite major advances in prevention strategies and initialized treatments in patients with high cardiovascular disease (CVD) risk factors. Atherosclerosis is a chronic silent disease, caused by the thickening or hardening of the inner arterial wall through the deposition of lipid plaques concomitant with localized inflammatory response, and it is the underlying pathological process of most cardiovascular diseases. Often, atherosclerotic patients present multiple vascular disorders: cerebrovascular disease, carotid artery and coronary heart disease (CHD), as well as peripheral artery disease (PAD).

Vulnerable plaques build up over time from the progressive participation of multiple immune cells with the release of pro-inflammatory cytokines, primarily C-reactive protein (CRP), interleukins 1, 6, and 8 (IL-1, IL-6, IL-8), and tumor necrosis factor-alpha (TNF-α) [1,2]. Plaque formation is initiated upon intimal vascular wall modifications by endothelial cell homeostasis loss attended by low density lipoprotein cholesterol (LDL-c) retention. Endothelial dysfunction predisposes vascular walls to vasoconstriction, lipid infiltration, leukocyte adhesion, platelet activation, and oxidative stress [3]. Monocytes, as well as macrophages, are recruited at the injury site by monocyte chemoattractant protein-1 (MCP-1), intercellular adhesion molecule-1 (ICAM-1), and vascular cell adhesion molecule 1 (VCAM-1) promoting foam cell formation. B and T lymphocytes, with LTh1, LTh2, and LTh17 subunits, release specific cytokines with pro and anti-inflammatory effects [4]. Furthermore, various inflammatory signaling pathways are activated, promoting fatty streak formation, an early sign of atherosclerosis. Shear stress mediates physiological processes such as endothelial, platelet, and leukocyte function and vascular development [5].

Atherothrombosis is influenced by shear stress response elements, which promote endothelial atherogenetic gene expression, and platelet adhesion molecule-1 (PECAM-1) with concomitant reduced inducible nitric oxide synthase (iNOS). Nitric oxide (NO) decreases in parallel with endothelial dysfunction, which promotes atherogenesis by leading directly to tissue oxidation and inflammation, activation of thrombogenic factors, and cell proliferation [4]. CVD major risk factors (smoking, hypertension, hyperlipidemia, hyperglycemia) also induce oxidative stress increase, which promotes pro-inflammatory cytokines synthesis, chemokines (MCP-1), and adhesion molecules (ICAM-1, VCAM-1) through nuclear factor κB (NF-κB) pathway induction. During atheroma plaque formation, fatty streaks may undergo some transitions to fibrous plaque through intimal growth and necrotic core development covered by a fibrous cap, the hallmarks of advanced atherosclerosis [4]. When the plaque develops with a lipid-rich core and a thin fibrous cap that may rupture, unstable plaques arise (coronary artery disease or CAD) and are susceptible to the rupture and thrombosis of the narrowed inflamed and vulnerable vessels. Atherosclerosis tends to develop at arterial branches, bends, and bifurcations, areas exposed to lower shear stress, therefore exposed to turbulent blood flow. Thrombosis might also arise in younger patients from vascular wall superficial erosion sites characterized by endothelial cell apoptosis on an underlying lipid-poor but proteoglycan and glycosaminoglycan-rich plaque structure [6]. Inflammation and lipid metabolism alteration contributes to plaque development at every stage, with a higher relevance in its last stage, as it promotes fibrous cap instability [7]. The mechanisms associated with unstable mixed or heterogenous ulcerated plaque development are of critical importance to define, and the reasons for their prevalence in diabetic individuals is the focus of this review.

### 1.2. Blood Sugar and Blocked Arteries: The Diabetes–Atherosclerosis Link

Metabolic disorders, a cluster of CVD-specific risk factors, including diabetes, are in a constant growth projection worldwide. Diabetes mellitus (DM) and atherosclerosis are multifactorial conditions that maintain the organism in a constant chronic inflammatory state with corresponding health repercussions. Among the main mechanisms of atherosclerosis induction in DM, the most significant are chronic inflammation, abnormal lipid metabolism, and secondary autoimmunity, which also induces a long-term state of systemic inflammation [8]. Patients with DM display a greater atherosclerotic plaque instability, with a consequently increased risk of myocardial infarction (MI) and death [9]. Whilst MI is the dominant cause of morbidity and mortality in diabetic individuals with atherosclerosis, CVD also associates strongly with manifestations such as MI, PAD, or stroke [10].

There is evidence suggesting that chronic refractory diabetic wounds are associated with the development of peripheral and systemic atherosclerosis, including features of plaque instability [11]. Hyperglycemia triggers the formation of advanced glycation end-products (AGEs), which damage the vascular endothelium, increase shear-stress-induced vascular modifications and oxidative stress, while promoting the classical inflammatory pathways’ activation (endothelial-to-mesenchymal transition (EndMT), monocyte infiltration, macrophage and foam cell formation, and finally vascular smooth muscle cell (VSMC) proliferation towards the plaque fibrous cap). This leads to insulin resistance, while sustaining a vicious circle through which the risk of diabetic complications increases [12,13]. Diabetic panvasculopathy involves microvascular dysfunction with endothelial injury, basement membrane thickening, erythrocyte and platelet microthrombus, as well as macrovascular dysfunction with endothelial injury and atherosclerosis. These changes usually end in an extensive macrovascular atherosclerosis disposed in a segmental distribution of numerous vascular branches [8]. Moreover, PAD, peripheral neuropathy, the chronic inflammatory state, and the altered cellular functions lead to the development of chronic diabetic wounds. One quarter of affected diabetic patients present with foot ulcers, infections, wound dehiscence, or chronic non-healing wounds sustained through chronic inflammation. In these types of lesions, macrophages produce an excess of pro-inflammatory cytokines, prolonging the inflammatory phase of healing, while neutrophils produce neutrophil extracellular traps (NETs) through oxidative stress, also contributing to excessive inflammation [13].

Diabetes-related foot ulceration (DFU), a “cardio–renal–metabolic–foot” axis in people with diabetes, cardiovascular, and renal disease, is a severe preventable complication caused by both peripheral neuropathy and PAD. Previous studies report that patients with DFU face more than a two-fold higher risk of ischemic heart disease and stroke compared to diabetic patients without ulcers [11]. Evidence suggests that renal dysfunction independently predicts DFU risk, with even moderate chronic kidney disease (CKD) (eGFR < 60 mL/min/1.73 m^2^) increasing the ulcer hazard by 1.85-fold and severe CKD (eGFR < 30 mL/min/1.73 m^2^) elevating it by 3.92-fold [14]. In addition, systemic inflammation (e.g., high CRP and/or IL-6 values) and infection from DFU drive endothelial dysfunction, accelerating atherosclerosis and renal damage. In addition, pre-existing CVD and CKD exacerbate DFU progression through microvascular impairment and immune dysregulation. Bidirectional relationships emerge where CVD/CKD promote DFU development, while DFU-associated inflammation worsens cardio-renal outcomes.

This integrated pathophysiology requires a holistic management targeting cardio–renal–metabolic health to mitigate DFU-related morbidity and mortality. Recent studies even suggest that, as EndMT plays a vital role in diabetes-accelerated atherosclerosis, targeting its signaling pathways might be a potential revolutionary therapeutic strategy. Future work is likely to focus upon and prioritize targeted delivery and combination therapies (through epigenetic regulators, transcription factors, TGF-β signaling, and miRNA modulation) to address residual cardiovascular risk in diabetic patients [10].

### 1.3. The Anatomy of an Inflammatory Marker (CRP) Linked to Cardiovascular Disease

A captivating protein with different roles in either physiological or pathophysiological states, CRP is an acute-phase protein, with implications in acute MI, stroke, infections, metabolic syndrome, obesity, heart failure, diabetes mellitus, autoimmune disorders, and various cancers [15,16]. Originally considered a non-specific marker of systemic inflammation, CRP is now recognized as a valuable independent predictor of future cardiovascular events, including atherothrombotic risk [17]. Under normal conditions, serum CRP levels range from 0 to 0.8–1.0 mg/dL, though values can vary by laboratory and are influenced by age, genetic background, and environmental factors. During inflammatory states, CRP concentrations can rise dramatically, by up to 25% or more, within 8 to 72 h. In cases of bacterial infection, CRP levels may increase up to 1000-fold, while tissue injury and malignancies may cause a 500-fold increase within 24 to 72 h [18,19].

Structurally, CRP exists primarily in two conformational isoforms: the native pentameric form (pCRP) and the monomeric dissociated form (mCRP). The native pCRP is composed of five identical subunits arranged in cyclic symmetry around a central pore and is stabilized by calcium ions, which enable binding to phosphatidylcholine on damaged cells or pathogens [20]. In contrast, mCRP, the product of pCRP dissociation, has a more direct role in local inflammation, activating platelets, leukocytes, and endothelial cells, and contributing to complement activation, angiogenesis, and thrombosis [21,22].

Importantly, while pCRP exerts more systemic, anti-inflammatory functions, mCRP is pro-inflammatory and is believed to contribute to plaque instability and progression of atherosclerosis [20,23,24,25]. The dissociation of pCRP to mCRP occurs at sites of tissue injury or vascular damage, mediated by factors such as acidic pH, oxidative stress, or contact with activated platelets and necrotic cells [26,27]. Clinically, high-sensitivity CRP (hs-CRP) assays are commonly used to detect circulating pCRP but are unable to differentiate or quantify mCRP. mCRP detection requires specialized techniques, such as immunofluorescence, sandwich enzyme-linked immunosorbent assay (ELISA), or monoclonal antibodies, for example, clone 8C8 used in flow cytometry [5,28]. This duality of CRP forms underscores its complex role in cardiovascular pathology, where it acts not only as a biomarker of risk but also as a mediator of vascular inflammation and thrombosis.

## 2. CRP and hs-CRP: Molecular Sentinels in Cardiometabolic Disease

Specific cardiovascular risk factors (age, gender, smoking status, hypertension, diabetes, dyslipidemia) cannot predict plaque instability or disruption [7], yet an accurate assessment is needed. Although high blood pressure and lipid accumulation have long been considered primary screening tools for cardiovascular risk, recent studies highlight the significant role of proteolytic activity, systemic inflammation, and their interplay in promoting atherogenesis and plaque instability [1,2]. As early as 2006, Krupinski et al. [29] observed overexpression of CRP in ulcerated, active, end-stage carotid plaques, indicative of their potential vulnerability. They suggested that local CRP synthesis may contribute to plaque neovascularization and an increased risk of hemorrhagic transformation. Subsequent research supports the role of CRP not only as a marker of systemic inflammation but also as a predictor of vulnerable atherosclerotic plaque activity [30]. hsCRP assays were developed to detect subtle changes in plasma CRP concentrations and have been used for over a decade in the prognosis of CVD. A growing body of evidence suggests that elevated hsCRP levels, typically defined as values above 2 mg/L, indicate a state of residual inflammatory risk. Major randomized controlled trials such as the Canakinumab Anti-Inflammatory Thrombosis Outcome Study (CANTOS), the Colchicine Cardiovascular Outcomes Trial (COLCOT), and low-dose colchicine (LoDoCo2) have demonstrated that reducing hsCRP below this threshold through anti-inflammatory therapy significantly lowers the incidence of major adverse cardiovascular events (MACE) by 23–29% [20].

Elevated serum CRP levels are closely associated with an increased risk of CAD and vascular-related mortality [23]. Persistently high baseline hs-CRP levels correlate with a 10% increase in cardiovascular events over the following decade [31]. High hs-CRP values are particularly concerning when combined with other risk factors, especially elevated LDL-C, and are linked to increased carotid plaque instability and thrombosis risk [7]. Ridker et al. [32] concluded that a composite assessment incorporating hs-CRP, LDL cholesterol, and lipoprotein(a) provides robust long-term predictive value for cardiovascular events, with lipid-rich plaques demonstrating a higher propensity for rupture [10]. Moreover, hs-CRP levels are significantly higher in patients with acute coronary syndrome (ACS) compared to those with stable CAD, underscoring its role in acute cardiac risk stratification [33]. Elevated hs-CRP in ACS patients is strongly predictive of adverse outcomes, including mortality and recurrence, emphasizing the importance of inflammation monitoring in CAD management.

Patients with type 2 diabetes mellitus (T2DM) are particularly vulnerable to CVD due to the convergence of hyperglycemia, dyslipidemia, and insulin resistance, which are the major factors that perpetuate chronic inflammation and oxidative stress [8,34]. Diabetes accelerates endothelial dysfunction and vascular remodeling via monocyte infiltration, macrophage activation, foam cell formation, and vascular smooth muscle cell (VSMC) proliferation, partly through shear stress and endothelial-to-mesenchymal transition (EndMT) mechanisms [35].

Inflammation, particularly low-grade and chronic, is a key driver linking diabetes to atherogenesis. Elevated levels of hs-CRP are consistently observed in diabetic patients, where they both reflect and perpetuate systemic inflammation. These elevated hs-CRP levels promote insulin resistance via induction of pro-inflammatory cytokines and innate immune activation [32]. Furthermore, diabetic individuals have a twofold higher risk of CHD, with increased plaque instability observed in over 60% of those affected (whilst 10-20% is considered the normal risk of vulnerability) [7,36]. Notably, hs-CRP is recognized as an independent prognostic factor for MACE in patients with three-vessel disease and T2DM [34]. However, it was noted by Stanimirovic et al. that, while hs-CRP elevation should be considered in risk assessments for T2DM, it remains an indirect indicator of disease progression [13]. There is also a significant association between genetic variation in the CRP gene and both elevated serum CRP levels and the increased risk of developing T2DM. Specifically, individuals carrying certain CRP single nucleotide polymorphisms (*SNPs*), particularly *rs1205*, also known as *CRP4*, exhibited higher baseline CRP concentrations, independent of traditional risk factors such as BMI or insulin resistance [37]. This finding supports the hypothesis that chronic low-grade inflammation, driven at least in part by genetic predisposition to elevated CRP, contributes to the pathogenesis of T2DM. These results suggest that CRP gene variants may act as early genetic markers for diabetes risk. The study contributes to the growing body of evidence positioning CRP as more than a passive biomarker and potentially a causal link in the inflammatory cascade leading to impaired glucose metabolism.

Epidemiological studies further validate the role of inflammation in diabetes onset and progression. The Women’s Health Study (2001) identified CRP as a powerful independent predictor of incident diabetes in over 27,000 women [38]. The West of Scotland Coronary Prevention Study (2002) [39] identified CRP as a predictor of diabetes development, independent of clinical features such as Body Mass Index (BMI), serum triglyceride levels or glucose concentrations, while the Monitoring of Trends and Determinants in Cardiovascular Disease (MONICA) Augsburg Cohort Study (2003) demonstrated a 2.7-fold increased diabetes risk in subjects with elevated CRP levels [40]. The Hisayama Study (2005) supported these findings in a Japanese cohort, highlighting CRP’s predictive role independently of obesity or insulin resistance [41]. The Chennai Urban Rural Epidemiology Study (CURES)-6 and CURES-38 studies expanded on these associations. CURES-6 showed that diabetic individuals, with or without CAD, had significantly elevated hs-CRP levels compared to non-diabetics, and hs-CRP was strongly associated with both diabetes and CAD [42]. CURES-38 further showed that worsening glucose intolerance increased the diabetes risk score and cardiovascular risk even in individuals with normal glucose tolerance [43].

Similarly, the Intervention Project on Cerebrovascular Diseases and Dementia in the District of Ebersberg (INVADE) study (2010) linked hyperglycemia and inflammation to early carotid atherosclerosis progression in a population of over 3500 subjects, with a significant interaction between hs-CRP and HbA1c in promoting increased carotid intima-media thickness (IMT) and vascular event risk [44]. The Action to Control Cardiovascular Risk in Diabetes (ACCORD) trial (2012) demonstrated that intensive glycemic control in 10,251 T2DM patients reduced hs-CRP levels, supporting inflammation reduction as a therapeutic target [45]. In more recent research, the Korean Genome and Epidemiology Study (KoGES) study (2019) involving nearly 23,000 individuals confirmed a positive association between higher CRP levels and incident T2DM, particularly in older populations and in the presence of obesity and hypertension [46]. Lastly, the Utrecht-based Second Manifestations of ARTerial disease (SMART) study (2021) found that low-grade inflammation, as indicated by hs-CRP, was associated with increased vascular and all-cause mortality in T2DM patients, though not with MI or stroke, suggesting its use as a mortality risk marker in high-risk diabetic populations [47]. These findings are summarized in Table 1.

This table summarizes major cohort and clinical studies investigating CRP and high-sensitivity CRP (hs-CRP) as predictive markers for incident diabetes, cardiovascular risk, and vascular complications. The studies vary in population size and focus, but collectively support the role of inflammation, as indicated by elevated CRP levels, in the pathogenesis and prognosis of T2DM and atherosclerosis. Some studies also highlight the interaction between metabolic factors (e.g., obesity, HbA1c) and inflammatory markers in advancing vascular disease.

Increasing evidence supports a central role for CRP, particularly hs-CRP, in linking inflammation, metabolic dysfunction, and CVD in individuals with T2DM. A study by Soinio et al. demonstrated that patients with T2DM and hs-CRP levels greater than 3 mg/L had a 1.6-fold higher risk of CHD death compared to those with lower levels, even after adjustment for other risk factors, supporting hs-CRP as an independent predictor of CHD mortality [48]. Similarly, in a large cohort of nearly 12,000 individuals, Pivato et al. found that elevated hs-CRP levels were associated with a higher incidence of major MACE, regardless of diabetes status [49]. The relationship between glycemic control and inflammation was further explored by Mojiminiyi et al., who observed progressively higher hs-CRP levels in angiographically documented CAD patients across three groups: normal glucose tolerance, impaired glucose regulation, and T2DM. The most significant elevations were seen in the group with both T2DM and CAD [50]. Supporting this, Bosevski et al. studied a cohort of 62 patients with T2DM and clinically confirmed atherosclerosis. They reported that CRP levels correlated positively with BMI, duration of diabetes, and the presence of metabolic syndrome, highlighting the involvement of CRP in atherosclerosis progression [51].

Further evidence links CRP to microvascular complications such as neuropathy, nephropathy, and retinopathy. Elevated hs-CRP has been specifically associated with the presence of neuropathy and impaired wound healing [15,52]. Studies also indicate that patients with diabetes who receive proper glycemic control through diet and treatment are at lower risk for vascular complications [53]. Mechanistic insights were provided by Dungu et al., who investigated the association between inflammation and insulin resistance in patients with community-acquired pneumonia. Their findings indicated that admission CRP levels were positively associated with insulin resistance, independent of blood glucose levels, suggesting that inflammation may precede hyperglycemia in this context. Furthermore, an increased BMI was linked to higher HOMA-IR values, indicating greater insulin resistance [54].

Public health data from the National Health and Nutrition Examination Survey (NHANES) involving 2693 adolescents revealed that more than 15% had elevated hs-CRP levels. Obesity was strongly associated with this systemic inflammation, suggesting a worrying future risk for CVD and diabetes complications in youth [55].

Collectively, these studies demonstrate that hs-CRP is consistently associated with insulin resistance, atherosclerosis, and adverse cardiovascular outcomes in T2DM and related conditions. Measuring hs-CRP may therefore offer valuable insights for early risk stratification and intervention in both clinical and public health settings. When considering the limitations of using CRP as a biomarker for inflammation in patients with atherosclerosis and diabetes, several important factors must be taken into account: CRP is a non-specific marker of inflammation, while it can arise from various conditions, including infections, autoimmune diseases or other inflammatory disorders. On the other hand, CRP levels might fluctuate due to a variety of factors, including acute infections, stress, or lifestyle changes such as diet or exercise. Although these studies focused on CRP in general, they did not specifically distinguish between its pentameric and monomeric forms. Further, more targeted research, is needed to elucidate the distinct roles of mCRP in insulin resistance and glucose metabolism, whilst it might be the key to this inflammatory biomarker on a large scale.

The most common hypothesis claims that atherosclerosis and diabetes arise from a “common soil”, where chronic inflammation plays a significant role in both conditions. Hs-CRP is considered an independent predictor of cardiovascular risk, with many studies indicating that hs-CRP levels correlate with the extent of atherosclerosis and that these are directly influenced by metabolic disease. The link between inflammation, atherosclerosis, and diabetes suggests that managing inflammation and monitoring CRP could be useful in prognosing the risk of cardiovascular complications in diabetic patients.

## 3. mCRP Prevalence in Diabetes-Driven Unstable Plaques

CRP, particularly its monomer mCRP, may play a significant role in the development and destabilization of atherosclerotic plaques. Unlike the circulating pCRP, mCRP is predominantly tissue-bound and exhibits potent pro-inflammatory properties. It promotes endothelial dysfunction, enhances leukocyte recruitment, and stimulates the release of matrix metalloproteinases (MMP), all of which contribute to plaque vulnerability.

In previous studies, mCRP was found bound to circulating microparticles in the blood of patients with carotid atherosclerotic plaque tissue [20,28], severe aortic valve stenosis [5], acute myocardial infarction (AMI) [22,28,56], and PAD [24,28], manifestations of systemic atherosclerosis. Recent studies highlight the fact that higher mCRP levels were associated with premature CAD, independent of hs-CRP levels or any other traditional risk factors [57].

Melnikov et al. [20] found no correlation between mCRP levels, hs-CRP levels, and IL-6 levels in two groups of patients with carotid subclinical atherosclerosis evaluated at baseline and at a 7-year follow-up, potentially indicating that hs-CRP alone fits the profile of being an adequate marker of inflammation, but the dissociation constant might truly be a sensitive determinant of risk and pathophysiological outcome. mCRP levels positively correlated though with von Willebrand Factor (vWF; a marker of activated endothelium)-measured levels, while hs-CRP positively correlated only with the IL-6 levels. Higher mCRP levels were also associated with a more pronounced plaque number and plaque height in patients with normal levels of traditional inflammatory biomarkers.

mCRP levels were measured in the blood plasma of patients with ACS and stable angina by Wang et al. [22]. Patients with AMI had significantly higher levels of mCRP compared to patients with unstable or stable angina and healthy controls, respectively. Zha et al. [56] observed a 12.7% left ventricular ejection fraction (LVEF) reduction and a significant larger infarction area at 1 week post-MI compared to baseline in mCRP-treated mice, concomitant with a significantly increased mCRP pro-inflammatory macrophage associated gene expression of IL-1β, TNF-α, CD40, and CD80. mCRP also induces platelet activation and transforming growth factor beta (TGFβ) expression on platelets [5]. Platelet aggregation was induced by mCRP and not pCRP, in a dose dependent manner. A prothrombotic effect of mCRP on platelet surfaces was shown, significantly further increasing platelet adhesion. In addition, mCRP increased the levels of P-selectin and CD63 on platelet surfaces, while the restriction of GPIIb-IIIa and CD36, significantly inhibited mCRP protein-induced activation [26]. According to Molins et al. [58], mCRP caused platelet adhesion and thrombus development, with an upregulation of P-selectin and CD36 during confocal microscopy analysis of thrombus growth under arterial flow, whereas pCRP granted anti-inflammatory effects by reducing the thrombus area.

Zeller et al. [5] also demonstrated by experiments in vitro, using an in vivo model, and on patients with stenotic aortic valves that after exposure to high shear stress rates, pCRP could dissociate into its pro-inflammatory monomers, correlating to the exposure time to shear stress. Interestingly, shear-dissociated mCRP did not bind to phosphocoline site, but rather unmasked extra C1q binding sites. In the absence of related-binding partners, the development of large aggregates was also observed. mCRP induced a significant increase in platelet TGFβ expression, intensifying fibrosis in aortic stenosis. They demonstrated that the deposition of mCRP on microvesicles of patients with severe aortic valve stenosis was reversible, as it significantly declined after a transcatheter aortic valve implantation (TAVI) procedure, and this could prove to be a protective mechanism.

mCRP effects on cultured human coronary artery endothelial cells are mediated via activation of the p38 mitogen-activated protein kinase (p38 MAPK) pathway, suggesting a significant contribution to endothelial cell injury development in atherosclerosis. Therefore, mCRP may be a potential signaling pathway regulator associated with ICAM-2, VCAM-1, VEGF, and cyclooxygenase-2 (COX-2) [59]. On blood preincubation with p38 MAPK and c-Jun N-terminal kinases (JNK) inhibitor, a significant reduction in mCRP’s effect on thrombocytogenesis was observed [26]. The JNK signaling pathway markedly inhibited the macrophage polarization development induced by mCRP in cultured cells, while NF-κB did not influence it. Responsible for the control of cardiac function recovery after AMI, macrophage phenotypes can be influenced by the JNK pathway; thus, mCRP increases and lengthens the period of inflammatory cell action in the infarcted area [56].

mCRP also binds strongly to endothelial and other cell-derived microparticles, inducing monocyte polarization towards M1 (pro-inflammatory/CD206^+^), in association with T-helper (TH2) derived production of interferon-gamma (IFNγ); furthermore, a strong correlation was found between LDL-c levels and mCRP bound to endothelial microparticles in individuals with PAD [28]. Since significantly higher levels of circulating EC microparticles are seen within the spectrum of diabetes, this combination with mCRP is potentially damaging to the microvascular milieu and a clear likely contributing factor to intimal plaque development and mixed or heterogenous inflammatory instability [60].

Similarly, mCRP induces monocyte aggregation, concomitant with increased expression of focal adhesion kinase (FAK), a key non-receptor protein tyrosine kinase in orchestrating cell movement and adhesion, and M1 phenotype transition. Moreover, ELISA indicated that mCRP-treated macrophages increased the expression of IL-8 and IL-1β, promoting a hyperinflammatory status and providing further supporting evidence that the local presence of mCRP could be contributing to plaque development over time [61].

### The RAGE-AGE-mCRP Axis in Diabetes

As stated previously, CRP dissociation into mCRP at sites of vascular and tissue damage causes vascular dysfunction via deregulated intracellular signaling and increased pro-inflammatory cytokine release. This promotes endothelial activation, macrophage M1 polarization, and chronic inflammation that impair diabetic wound repair and exacerbate vascular damage. Chronic hyperglycemia and associated metabolic disturbances in diabetes promote oxidative stress, endothelial dysfunction, and low-grade inflammation, creating an ideal environment for pCRP dissociation and mCRP accumulation in vascular tissues. Elevated glucose levels increase the formation of AGEs, which interact with RAGE on endothelial and immune cells, amplifying inflammatory signaling pathways that synergize with mCRP-mediated effects [62].

mCRP and AGEs also synergize by mutually enhancing endothelial dysfunction and inflammation. mCRP promotes RAGE expression and signaling, thereby amplifying AGE-mediated oxidative stress and cytokine production, whereas CRP (undefined type) was able to upregulate the expression of RAGE receptors when applied to human saphenous vein endothelial cells in vitro [63]. Conversely, AGE–RAGE activation facilitates pCRP dissociation into mCRP at sites of vascular injury, perpetuating a feed-forward loop of inflammation. Synergistically, both mCRP and AGEs influence complement activation [64,65]. mCRP binds complement regulators like C4b-binding protein (C4BP) and modulates classical complement pathway activation by recruiting C1q to damaged cells. AGEs can also activate complement via modified proteins, contributing to chronic vascular inflammation. mCRP also binds integrins αvβ3 and α4β1 on monocytes and endothelial cells, activating protein kinase B (AKT) signaling and promoting chemotaxis and inflammatory responses. AGE-modified proteins similarly interact with integrins and RAGE, enhancing leukocyte adhesion and transmigration into the vascular wall (Figure 1).

This diagram illustrates the mechanistic links contributing to vascular inflammation and plaque instability in diabetes. Hyperglycemia induces the formation of advanced glycation end products (AGEs), which activate the receptor for AGEs (RAGE) on endothelial and immune cells. This amplifies oxidative stress and cytokine release and promotes dissociation of pentameric CRP (pCRP) into its monomeric form (mCRP) at sites of vascular injury. mCRP, in turn, enhances RAGE signaling and promotes endothelial activation, monocyte adhesion, and M1 macrophage polarization. These effects increase the expression of inflammatory mediators (e.g., IL-1β, IL-8, TNF-α), matrix metalloproteinases, and adhesion molecules (VCAM-1, ICAM-1), thereby weakening the fibrous cap and accelerating plaque vulnerability. LDL cholesterol acts synergistically by contributing to foam cell formation and complement activation. The feed-forward loop between mCRP and the RAGE–AGE axis promotes chronic inflammation, platelet activation, and integrin-mediated signaling (e.g., via αvβ3, α4β1), which together foster the development of unstable rupture-prone atherosclerotic plaques in individuals with diabetes.

## 4. RAGE Signaling Pathway Aggravates Diabetes-Mediated Vascular Calcification

The AGE–RAGE complex exerts direct toxic effects on pancreatic β-cells, and it also impairs glucose-induced insulin secretion, further triggering diabetic complications. Activation of the AGE–RAGE axis leads to increased cholesterol intake in macrophages and stimulates the proliferation and migration of arterial SMCs, leading to the thickening of the arterial intima and plaque formation, as well as to endothelial cell dysfunction and dysregulation of the vascular constriction–dilation mechanism [66].

Emerging evidence supports a mechanistic link between the RAGE–CRP axis and vascular calcification, particularly in the context of diabetes. Hyperglycemia and mineral imbalances such as hyperphosphatemia and hypercalcemia, due to diabetes, place these patients at a higher risk for cardiovascular disease. VSMCs’ calcification is AGE–RAGE signaling pathway-dependent [67]. Chronic elevated glucose values promote the formation of AGEs, which engage the receptor for AGEs (RAGE), leading to oxidative stress and activation of pro-inflammatory transcription factors such as NF-κB. Simultaneously, mCRP (generated from the dissociation of pentameric CRP at sites of vascular injury) enhances RAGE expression and signaling [68]. This synergistic interaction amplifies inflammatory oxidative stress, ECM breakdown, and cytokine release and upregulates osteogenic transcription factors like Runx2 and Bone morphogenetic protein 2 (BMP2) in VSMCs, driving their phenotypic switch and contributing to calcific remodeling [69]. The mCRP–RAGE axis thus forms a pathogenic feed-forward loop that accelerates plaque instability and vascular calcification (unstable-spotted microstructure in mixed plaques), highlighting a potential therapeutic target in diabetes-related CVD (Figure 2).

The diagram illustrates how hyperglycemia-induced AGEs activate RAGE, leading to oxidative stress, NF-κB activation, and inflammatory cytokine release. Concurrently, monomeric CRP (mCRP) amplifies RAGE signaling and promotes the expression of osteogenic markers (Runx2, BMP2) in vascular smooth muscle cells (VSMCs), contributing to vascular calcification. This feed-forward loop enhances endothelial dysfunction, inflammation, and plaque instability in diabetic vasculature.

In the broader sense, in all vascular aging, which is also prevalent in general, arteriosclerosis (not only in the major arteries) and increased arterial stiffness are markers for the progression of CVD in patients with T2DM by facilitating unstable disease phenotypes and raising mortality risk. As outlined by Climie et al. (2023), vascular aging is marked by structural deterioration (e.g., loss of elastin, collagen cross-linking due to AGEs) and functional decline, including reduced arterial compliance and endothelial dysfunction [70]. Specifically, in T2DM, hyperglycemia and dyslipidemia accelerate these changes, promoting plaque vulnerability with characteristically larger necrotic cores and thinner fibrous caps that predispose to rupture and adverse cardiovascular outcomes [10]. Monitoring indicators such as the pulse wave velocity (PWV), carotid–femoral PWV, and imaging-based measures (e.g., coronary artery calcification scores or supra-stiffness) should enable early detection of subclinical vascular disease. These non-invasive assessments are strongly predictive of all-cause and cardiovascular mortality, but not necessarily MI or stroke, mirroring findings from cohorts like SMART and studies of hs-CRP. In relation to this review, Mozos et al. (2019) showed that elevated hs-CRP levels were significantly associated with increased arterial stiffness parameters, including augmentation index (AIx) and PWV, in hypertensive patients, supporting its use as an early indicator of vascular aging and dysfunction within susceptible patient cohorts [71].

Such findings emphasize that arterial aging and stiffness function as robust prognostic markers and highlight the importance of integrating vascular aging assessments in routine screening protocols for high-risk diabetic populations.

## 5. Conclusions

The association between persistent inflammation, as characterized by elevated plasma CRP levels, and chronic diseases has attracted significant clinical interest. Inflammation in CVD and diabetes is closely interconnected, with each condition capable of exacerbating the other, thereby increasing morbidity and mortality. CRP has established itself as a reliable, accessible, and independent predictor of vascular health and atherosclerosis progression. mCRP, the dissociated form of circulating pCRP, plays a crucial role in the development and destabilization of atherosclerotic plaques through its potent pro-inflammatory effects. Predominantly tissue-bound, mCRP promotes endothelial dysfunction, leukocyte recruitment, platelet activation, and MMP release factors that contribute to plaque vulnerability and thrombotic events. Clinical studies have reported elevated mCRP levels in patients with various cardiovascular conditions, including AMI, PAD, and aortic valve stenosis, often independently of traditional inflammatory markers such as hs-CRP. Mechanistically, mCRP activates key intracellular signaling pathways, including p38 MAPK in endothelial cells, and drives macrophage polarization toward a pro-inflammatory M1 phenotype, thereby sustaining chronic vascular inflammation and promoting plaque instability. Hence, the somewhat non-specific nature and susceptibility to confounding factors (acute infections, obesity, smoking, physical inactivity, etc.) of pCRP, can limit its reliability as an independent predictor of diabetes-related atherosclerosis. These confounders can elevate CRP levels irrespective of underlying vascular pathology, making it difficult to isolate its contribution to the atherosclerotic process specifically in diabetic populations. Although elevated CRP is consistently associated with cardiovascular risk, its predictive value in diabetes-related atherosclerosis should be interpreted in conjunction with other vascular aging markers and risk factors, rather than as a standalone biomarker. Hence there is a need for multi-modal approaches to risk stratification and more specifically the enhanced utilization of plasma mCRP measurements, where the weight of evidence as to its greater specificity is growing exponentially.

In diabetes, the pathogenic impact of mCRP is significantly amplified. Chronic hyperglycemia, oxidative stress, and low-grade inflammation associated with diabetes enhance mCRP accumulation and activity within vascular tissues. Elevated glucose levels increase the formation of AGEs, which bind to their receptor (RAGE) and synergize with mCRP to exacerbate endothelial dysfunction and vascular inflammation. This AGE–RAGE–mCRP axis establishes a vicious cycle of tissue injury, complement activation, and enhanced leukocyte adhesion, accelerating the progression and destabilization of atherosclerotic plaques. Given the high prevalence of vascular complications in individuals with diabetes, understanding and targeting this pathological interplay is essential for developing effective therapies to reduce cardiovascular risk and improve outcomes in diabetic patients.

The key points raised within this review include the unique emphasis of CRP/mCRP not only as a static biomarker of systemic inflammation but also a dynamic indicator of vascular aging and arterial stiffness. Its further alignment with advanced vascular assessments (e.g., pulse wave velocity, intima-media thickness) to improve early prediction of unstable cardiovascular phenotypes supports the use of CRP/mCRP as an early detector of preclinical endothelial dysfunction and vascular remodeling, preceding overt atherosclerosis or ischemic events. Finally, this review highlights CRP’s role in destabilizing vascular tissue via immune signaling, smooth muscle cell activation, and endothelial dysfunction, linking chronic low-grade inflammation to increased arterial vulnerability and eventually plaque instability, especially in patients with diabetes or metabolic syndrome. We propose that routine monitoring of inflammatory and vascular aging markers could better capture dynamic cardiovascular risk, particularly in high-risk groups, than traditional static risk calculators alone.

## 6. Future Directives: Targeting the mCRP–RAGE Axis: A Therapeutic Opportunity in Diabetes-Associated Plaque Instability

Growing evidence supports the pathogenic association of mCRP and RAGE as a critical driver of vascular inflammation and plaque destabilization, particularly in the context of DM. Chronic hyperglycemia, characteristic of diabetes, promotes the formation of AGEs, which engage RAGE on endothelial and immune cells, initiating a cascade of oxidative stress and NF-κB-mediated pro-inflammatory signaling. Concurrently, pCRP undergoes dissociation into its monomeric form at sites of endothelial injury, where mCRP amplifies local inflammation, promotes M1 macrophage polarization, induces MMP release, and enhances platelet activation, all of which are key contributors to atherogenesis and plaque vulnerability.

Importantly, a feed-forward loop appears to exist whereby mCRP upregulates RAGE expression and signaling, while RAGE activation facilitates the conversion of circulating pCRP to its pro-inflammatory monomeric form. This synergistic relationship exacerbates endothelial dysfunction, leukocyte adhesion, and vascular remodeling, thereby accelerating the development of unstable atherosclerotic plaques in diabetic patients. Preclinical studies have demonstrated that mCRP activates key intracellular pathways, including p38 MAPK, JNK, and AKT, further propagating the inflammatory response and promoting adverse vascular remodeling. Targeted intervention at the level of the mCRP–RAGE axis, using RAGE antagonists, mCRP inhibitors, or agents that prevent pCRP dissociation, holds significant promise for reducing vascular inflammation, stabilizing plaques, and ultimately lowering cardiovascular event risk in diabetes. As such, this axis represents a compelling therapeutic target in the prevention and management of diabetes-associated cardiovascular complications.

In patients with type 2 diabetes, metabolic syndrome, or early vascular aging, chronic low-grade inflammation and oxidative stress drive progressive endothelial dysfunction and arterial stiffening, which are hallmarks of increased cardiovascular risk. The biomolecular pathways involving the AGE–RAGE axis and mCRP are central to this process. AGE–RAGE activation promotes NF-κB signaling, oxidative stress, and vascular remodeling, while mCRP acts locally at sites of vascular injury to amplify leukocyte recruitment, cytokine production, and endothelial activation. In this high-risk population, a therapeutic strategy that targets both pathways could attenuate vascular inflammation, reduce arterial stiffness, and ultimately lower the risk of cardiovascular events such as myocardial infarction or stroke. A combined therapy might include a RAGE antagonist or soluble RAGE (sRAGE) decoy receptor to neutralize AGE signaling, alongside an agent that either prevents CRP dissociation (e.g., phosphocholine mimetics) or neutralizes mCRP using isoform-specific antibodies. These may realistically become available as therapeutics, as Zeller et al. recently developed a novel phosphocholine-mimetic, C10M, which selectively bound to pCRP to inhibit its conformational transition into the pro-inflammatory intermediate pCRP* and mCRP. In vitro, C10M effectively blocked pCRP’s binding to activated platelets and endothelial cells, markedly reduced monocyte–platelet aggregation, downregulated adhesion molecule expression (ICAM-1, VCAM-1), and inhibited ROS generation and NET formation, while preserving CRP’s pathogen-opsonizing activity. Activity was also seen in vivo, where administration of C10M in murine models of renal ischemia–reperfusion injury and hindlimb allograft rejection significantly diminished the tissue deposition of inflammatory CRP isoforms, reduced monocyte infiltration, improved organ function, and prevented graft failure [72].

Adjunctive agents such as statins, metformin, or SGLT2 inhibitors, which have pleiotropic effects on both inflammation and glycation—could enhance the anti-inflammatory effect. Together, these interventions aim to interrupt the AGE–RAGE–mCRP inflammatory loop, stabilize vascular function, and shift management from reactive to preventative care in individuals most vulnerable to unstable CVD.

In summary, CRP/mCRP demonstrates significant value in the early detection and risk stratification of CVD, particularly among high-risk populations such as patients with T2D. Elevated levels of hsCRP reflect low-grade systemic inflammation, a key driver of vascular aging, endothelial dysfunction, and atherogenesis, pathophysiological processes that often precede clinical cardiovascular events. In diabetic individuals, who typically experience accelerated vascular aging and heightened cardiometabolic risk, CRP serves as a sensitive and accessible biomarker that correlates with increased arterial stiffness, plaque instability, and mortality risk, even in the absence of overt MI or stroke. As such, measuring hsCRP (and possibly in the future mCRP within the plasma) can enable clinicians to identify subclinical disease earlier, guide preventive interventions, and improve risk stratification in this vulnerable population. This makes CRP a practical and cost-effective tool for integration into cardiovascular risk monitoring protocols, particularly in the context of chronic inflammatory comorbidities like diabetes.

## 7. Limitations

Despite growing recognition of mCRP as a potent pro-inflammatory mediator in vascular and inflammatory diseases, the clinical evidence supporting its role in disease progression and cardiovascular outcomes remains limited. Much of the current understanding is derived from in vitro studies, animal models, or small observational cohorts, which, while mechanistically informative, fall short of establishing mCRP as a validated clinical biomarker. A major obstacle is the technical difficulty in reliably detecting and quantifying mCRP in human samples. Unlike the circulating pCRP, mCRP is typically tissue-bound and locally active, and there is a lack of widely available standardized assays capable of distinguishing between CRP isoforms. Moreover, the temporal relationship between mCRP expression and clinical events remains unclear; it has not yet been established whether mCRP acts as a driver of disease pathology or merely accumulates at sites of tissue damage. This ambiguity is further complicated by heterogeneity in disease models, as mCRP may exert context-specific effects across cardiovascular, autoimmune, and neurodegenerative conditions. Notably, no interventional studies have yet evaluated whether directly targeting mCRP or inhibiting its formation translates into improved outcomes. To address these gaps, future research should prioritize the development of sensitive isoform-specific assays for clinical use, alongside large prospective cohort studies to examine mCRP dynamics in relation to vascular aging, plaque vulnerability, and long-term cardiovascular risk. Interventional trials investigating pharmacological strategies that block CRP dissociation or neutralize mCRP’s biological activity are also warranted. Ultimately, clarifying the causal role of mCRP in human disease and validating its utility as a biomarker or therapeutic target could significantly enhance risk stratification and personalized treatment in cardiovascular and inflammatory disorders.

Whilst we found no well-documented reports that contradict a pathogenic role for mCRP in CVD or diabetes-associated vascular complications, most available supporting evidence originates still from in vitro studies, animal models, and small-scale clinical investigations consistently supporting the view that mCRP promotes vascular inflammation, endothelial dysfunction, and thrombotic activity at sites of tissue injury (whilst suffering from the limitation of a lack of translation in human study). Hence, the current body of evidence is limited by a lack of large prospective human studies, raising the possibility that the role of mCRP could be more nuanced than presently understood. Additionally, the absence of standardized isoform-specific assays complicated, until recently, accurate measurement in clinical settings and may have obscured potentially neutral or context-specific effects. While no studies to date have demonstrated a protective or non-pathogenic role for mCRP in CVD, future research, including longitudinal cohort studies, genetic analyses, and interventional trials, is needed to definitively confirm its causal role or to challenge prevailing assumptions.

Regarding type-1 diabetes, since vascular dysfunction and inflammation are also key features of this disease we can extrapolate that CRP and particularly mCRP could be implicated in the early vascular complications observed in T1D. While most data focus on T2D and CVD, the pro-inflammatory properties of mCRP, such as its ability to promote leukocyte recruitment, endothelial activation, and oxidative stress, may similarly accelerate vascular aging in T1D, which is characterized by chronic autoimmune inflammation and heightened cardiovascular risk even at a young age.

Furthermore, because children and young adults with T1D often show early endothelial impairment and increased arterial stiffness, the inflammatory cascade mediated by CRP/mCRP could serve as a common effector pathway linking autoimmune activity to vascular damage. However, the absence of direct in vivo studies or clinical trials examining CRP isoforms in T1D populations highlights a significant knowledge gap. Future research should investigate whether mCRP accumulates in pancreatic or vascular tissues in T1D, whether its levels correlate with disease duration or microvascular complications, and whether modulating CRP dissociation (e.g., via agents like C10M) could offer therapeutic benefit in T1D-related vascular dysfunction. In summary, while extrapolation is theoretically sound based on shared pathophysiological pathways, dedicated studies in T1D are essential to confirm causality and clinical relevance.

## Figures and Tables

**Figure 1 ijms-26-06855-f001:**
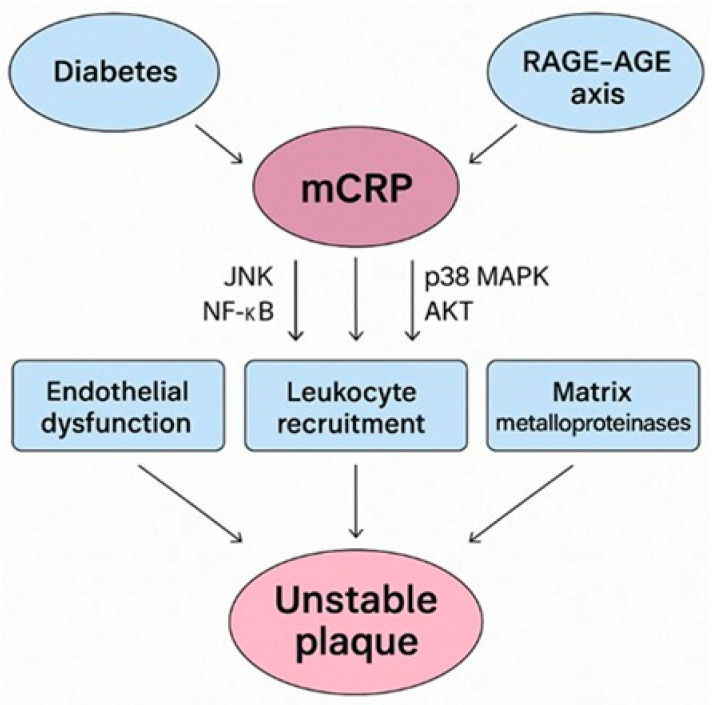
Pathophysiological interplay between mCRP, the RAGE–AGE axis, and diabetes-associated unstable plaque development.

**Figure 2 ijms-26-06855-f002:**
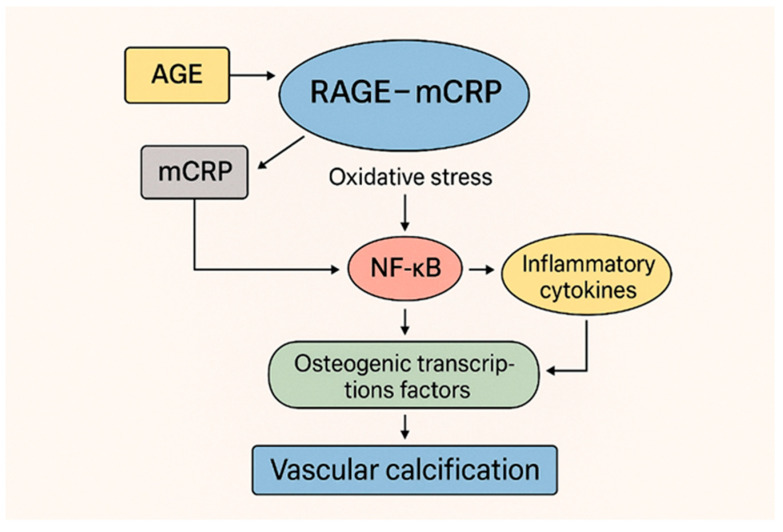
Interplay between mCRP, RAGE–AGE signaling, and vascular calcification in diabetes.

**Table 1 ijms-26-06855-t001:** Key population studies examining the association between C-reactive protein (CRP) and the risk of type 2 diabetes mellitus (T2DM), cardiovascular disease (CVD), and related outcomes.

Study	Year of Publication	Cohort	Developed Diabetes	Association	Conclusion
The Women’s Health Study [38]	2001	27,628	188	CRP–IL-6–diabetes	Elevated levels of CRP and IL-6 predict the development of T2DM
The West of Scotland Coronary Prevention Study [39]	2002	5245	127	CRP–T2DM	CRP is an important independent predictor of diabetes development Low grade inflammation is connected to the diabetes pathogenesis
MONICA Augsburg Cohort Study [40]	2003	2052	101	CRP–diabetes	High CRP levels correspond to a 2.7 times higher risk of diabetes development Low grade inflammation is associated with increased T2DM risk
Hisayama study [41]	2005	1759	131	CRP–diabetes	Elevated CRP concentrations are a significant diabetes predictor, independent of obesity/insulin resistance
CURES-6 study [42]	2005	26,001	150	CRP–diabetes–CAD Body fat–diabetes–CAD	Diabetics with and without CAD had significantly higher CRP levels Hs-CRP levels increased with body fat and HbA1c increase Hs-CRP was strongly associated with CAD and diabetes
CURES-38 study [43]	2007	2350	146	Diabetes risk score–glucose intolerance	The diabetes risk score increases with increasing glucose intolerance The diabetes risk score is an effective indicator of metabolic syndrome and cardiovascular risk
INVADE study [44]	2010	3534	882	IMT progression–HbA1c and hsCRP	Hyperglycemia and inflammation are associated with an advanced early atherosclerosis progression and an increased risk of new vascular events in diabetic/nondiabetic subjects
The Action to Control Cardiovascular Risk in Diabetes (ACCORD) trial [45]	2012	10,251	562	Intensive glycemic control–lower hs-CRP	Intensive glycemic control leads to hs-CRP adjustment Adjusting BMI/waist circumference led to lower hs-CRP values
KoGES study [46]	2019	22,946	278	CRP, obesity, hypertension–T2DM	CRP is an independent risk determinant, or in combination with obesity and hypertension, of diabetes
Second Manifestations of ARTerial disease (SMART) [47]	2021	1679 with diabetes	650 with CV events	hs-CRP not associated with myocardial infarction, stroke, or vascular disease in T2DM	Low-grade inflammation (measured through hs-CRP) is an independent risk factor for vascular and all-cause mortality, but not for CV events in high risk T2DM subjects

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
