# Peer review of "The Pathogenic Role of C-Reactive Protein in Diabetes-Linked Unstable Atherosclerosis"

_ijms, 2025, doi:10.3390/ijms26146855_

Round 1
Reviewer 1 Report
Comments and Suggestions for Authors
This review explores the growing recognition of monomeric C-reactive protein (mCRP) as a key player in the development of diabetes mellitus and cardiovascular disease (CVD). Although CRP has traditionally been used as a marker of systemic inflammation and cardiovascular risk, its monomeric form demonstrates unique and powerful pro-inflammatory actions, particularly at the tissue level. mCRP actively contributes to vascular damage by inducing endothelial dysfunction, attracting leukocytes, activating platelets, and altering macrophage behavior—factors that collectively drive the formation of unstable atherosclerotic plaques. In individuals with diabetes, these harmful effects are intensified by metabolic imbalances, sustained high blood sugar, and increased levels of advanced glycation end products (AGEs). The interaction between mCRP and the AGE-RAGE axis further amplifies oxidative stress and inflammation in the vasculature, accelerating atherosclerotic progression and heightening cardiovascular risk in diabetic patients. This review underscores the potential of mCRP not only as a diagnostic marker but also as a therapeutic target, and it calls for further investigation into interventions that can interrupt the mCRP–AGE-RAGE inflammatory loop to enhance cardiovascular outcomes in vulnerable populations. The conclusions are consistent with the evidence and arguments presented. The references are appropriate. The tables and figures are satisfactory. I recommend accepting the manuscript.
Reviewer 2 Report
Comments and Suggestions for Authors
The present paper aimed to review the crosstalk between cardiovascular disorders and diabetes mellitus pathophysiology, emphasizing the role of CRP and mCRP. The paper demonstrates that CRP is not only a predictive biomarker for cardiovascular events but may also actively contribute to atherogenesis by promoting inflammation within the vessel wall.
A few changes are needed, as follows:
Please explain every abbreviation before using it!
Table 1. In Population, please mention what type of patients were included.
Line 368: Please explain what “LTh1 polarization modulation” means!
It is impossible to talk about vascular health without mentioning not just atherosclerosis, but also arteriosclerosis, vascular aging and arterial stiffness, also related to cardiovascular events (Vascular ageing: moving from bench towards bedside. Eur J Prev Cardiol. 2023 Aug 21;30(11):1101-1117. doi: 10.1093/eurjpc/zwad028. Erratum in: Eur J Prev Cardiol. 2023 Aug 21;30(11):1165. doi: 10.1093/eurjpc/zwad134). It is also worth mentioning that high-sensitivity C-reactive protein is a sensitive predictor of early arterial (vascular) aging (Links between High-Sensitivity C-Reactive Protein and Pulse Wave Analysis in Middle-Aged Patients with Hypertension and High Normal Blood Pressure. Dis Markers. 2019 Jul 17;2019:2568069. doi: 10.1155/2019/2568069).
Please also mention that a genetic variant in the human CRP locus was identified, associated with a high serum CRP and an increased risk of diabetes (Genetic variation, C-reactive protein levels, and incidence of diabetes. Diabetes. 2007 Mar;56(3):872-8. doi: 10.2337/db06-0922).
Please highlight the value of CRP for early detection and risk stratification in high-risk patients groups, especially diabetic patients.
Reviewer 3 Report
Comments and Suggestions for Authors
The manuscript appears to emphasize the interaction between mCRP and the AGE-RAGE signaling, in the context of diabetes, but does the manuscript provide a sufficient association to novel mechanistic insights to make the results interesting at this time (beyond what we already know in the current literature)?
Acknowledging the limitations of current assays to differentiate between pCRP and mCRP in vivo, how trustworthy are the relationships between mCRP versus disease outcomes or the studies cited enough to support the mCRP-specific measurements in their studies?
The review discusses the potential for targeting the mCRP-RAGE axis as therapy. Are we to expect generalized examples of success from in vivo or clinical trials targeting this suggested axis, or is this still too hypothetical?
The authors acknowledge that CRP is a non-specific inflammatory marker that is impacted by many other confounders (e.g infection, lifestyle); therefore, in what ways does this limit the confidence in the conclusions regarding CRP to make independent predictions regarding diabetes-related atherosclerosis?
Many of the referenced population studies regarding CRP do not distinguish between pCRP and mCRP. Does this limit the strength of the argument that mCRP is, in fact, the important pathogenic factor regarding plaque instability as it relates to diabetes?
High-sensitivity CRP (hs-CRP) assays cannot differentiate mCRP from pCRP, so how can the review confidently state that mCRP is a superior biomarker or therapeutic target based on studies measuring hs-CRP?
Does the review sufficiently review the studies that contradict or fail to support a pathogenic role for mCRP, or is there inherent selective citation bias?
Are there any validated, safe, and effective pharmacological agents specifically targeting mCRP in clinical settings, or is this purely theoretical? What are the potential risks of targeting mCRP given its possible physiological roles?
Many mechanistic insights are derived from in vitro studies or animal models. What is the transferability of these findings to human pathophysiology, especially within the multifactorial context of patients with diabetes?
The authors do not discuss the distinction between Type 1 and Type 2 diabetes in most mechanistic discussions. As the pathophysiological landscapes for both diseases are distinct, can conclusions be extrapolated to these populations?
Round 2
Reviewer 3 Report
Comments and Suggestions for Authors
The paper can be accepted in its present form.